# Dynamic Scale Inference by Entropy Minimization

## Abstract

Given the variety of the visual world there is not one true scale for recognition: objects may appear at drastically different sizes across the visual field. Rather than enumerate variations across filter channels or pyramid levels, dynamic models locally predict scale and adapt receptive fields accordingly. The degree of variation and diversity of inputs makes this a difficult task. Existing methods either learn a feedforward predictor, which is not itself totally immune to the scale variation it is meant to counter, or select scales by a fixed algorithm, which cannot learn from the given task and data. We extend dynamic scale inference from feedforward prediction to iterative optimization for further adaptivity. We propose a novel entropy minimization objective for inference and optimize over task and structure parameters to tune the model to each input. Optimization during inference improves semantic segmentation accuracy and generalizes better to extreme scale variations that cause feedforward dynamic inference to falter.

## 1 Introduction

The world is infinite in its variations, but our models are finite. While inputs differ in many dimensions and degrees, a deep network is only so deep and wide. To nevertheless cope with variation, there are two main strategies: static enumeration and dynamic adaptation. Static enumeration defines a set of variations, processes them all, and combines the results. For example, pyramids enumerate scales (Burt & Adelson, 1983; Kanazawa et al., 2014) and group-structured filters enumerate orientations (Cohen & Welling, 2017). Dynamic adaptation selects a single variation, conditioned on the input, and transforms processing accordingly. For example, scale-space search (Lindeberg, 1994; Lowe, 2004) selects a scale transformation from input statistics and end-to-end dynamic networks select geometric transformations (Jaderberg et al., 2015; Dai et al., 2017), parameter transformations (De Brabandere et al., 2016), and feature transformations (Perez et al., 2017) directly from the input. Enumeration and adaptation both help, but are limited by computation and supervision, because the sets enumerated and ranges selected are bounded by model size and training data.

Deep networks for vision exploit enumeration and adaptation, but generalization is still limited. Networks are enumerative, by convolving with a set of filters to cover different variations then summing across them to pool the variants (LeCun et al., 1998; Krizhevsky et al., 2012; Zeiler & Fergus, 2014). For scale variation, image pyramids (Burt & Adelson, 1983) and feature pyramids (Shelhamer et al., 2017; Lin et al., 2017) enumerate scales, process each, and combine the outputs. However, static models have only so many filters and scales, and may lack the capacity or supervision for the full data distribution. Dynamic models instead adapt to each input (Olshausen et al., 1993). The landmark scale invariant feature transform (Lowe, 2004) extracts a representation adapted to scales and orientations predicted from input statistics. Dynamic networks, including spatial transformers (Jaderberg et al., 2015) and deformable convolution (Dai et al., 2017), make these predictions and transformations end-to-end. Predictive dynamic inference is however insufficient: the predictor may be imperfect in its architecture or parameters, or may not generalize to data it was not designed or optimized for. Bottom-up prediction, with only one step of adaptation, can struggle to counter variations in scale and other factors that are too large or unfamiliar.

To further address the kinds and degrees of variations, including extreme out-of-distribution shifts, we devise a complementary third strategy: unsupervised optimization during inference. We define an unsupervised objective and a constrained set of variables for effective gradient optimization. Our novel inference objective minimizes the entropy of the model output to optimize for confidence. The variables optimized over are task parameters for pixel-wise classification and structure parameters

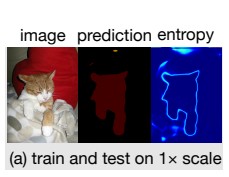 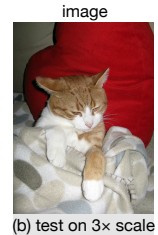 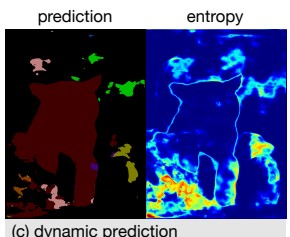 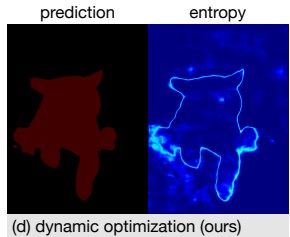

Figure 1: Generalization across scale shifts between training and testing conditions is difficult. Accuracy is high and prediction entropy is low for training and testing at the same scale (left). Accuracy drops and entropy rises when tested at 3x the training scale, even when the network is equipped with dynamic receptive fields to adapt to scale variation (middle). Previous approaches are limited to one-step, feedforward scale prediction, and are unable to handle a 3x shift. In contrast our iterative gradient optimization approach is able to adapt further (right), and achieve higher accuracy by minimizing entropy with respect to task and scale parameters.

for receptive field adaptation, which are updated together to compensate for scale shifts. This optimization functions as top-down feedback to iteratively adjust feedforward inference. In effect, we update the trained model parameters to tune a custom model for each test input.

Optimization during inference extends dynamic adaptation past the present limits of supervision and computation. Unsupervised optimization boosts generalization beyond training by top-down tuning during testing. Iterative updates decouple the amount of computation, and thus degree of adaptation, from the network architecture. Our main result is to demonstrate that adaptation by entropy optimization improves accuracy and generalization beyond adaptation by prediction (see Figure 1), which we show for semantic segmentation by inference time optimization of a dynamic Gaussian receptive field model (Shelhamer et al., 2019) on the PASCAL VOC (Everingham et al., 2010) dataset.

## 2 ITERATIVE DYNAMIC INFERENCE BY UNSUPERVISED OPTIMIZATION

Our approach extends dynamic scale inference from one-step prediction to multi-step iteration through optimization. For optimization during inference, we require an objective to optimize and variables to optimize over. Lacking task or scale supervision during inference, the objective must be unsupervised. For variables, there are many choices among parameters and features. Our main contribution is an unsupervised approach for adapting task and structure parameters via gradient optimization to minimize prediction entropy.

Note that our *inference* optimization is distinct from the *training* optimization. We do not alter training in any way: the task loss, optimizer, and model are entirely unchanged. In the following, optimization refers to our inference optimization scheme, and not the usual training optimization.

To optimize inference, a base dynamic inference method is needed. For scale, we choose local receptive field adaptation (Dai et al., 2017; Zhang et al., 2017; Shelhamer et al., 2019), because scale varies locally even within a single image. In particular, we adopt dynamic Gaussian receptive fields (Shelhamer et al., 2019) that combine Gaussian scale-space structure with standard "free-form" filters for parameter-efficient spatial adaptation. These methods rely on feedforward regression to infer receptive fields that we further optimize.

Figure 2 illustrates the approach. Optimization is initialized by feedforward dynamic inference of Gaussian receptive fields (Shelhamer et al., 2019). At each following step, the model prediction and its entropy are computed, and the objective is taken as the sum of pixel-wise entropies. Model parameters are iteratively updated by the gradient of the objective, resulting in updated predictions and entropy. Optimization of the parameters for the Gaussian receptive fields is instrumental for adapting to scale.

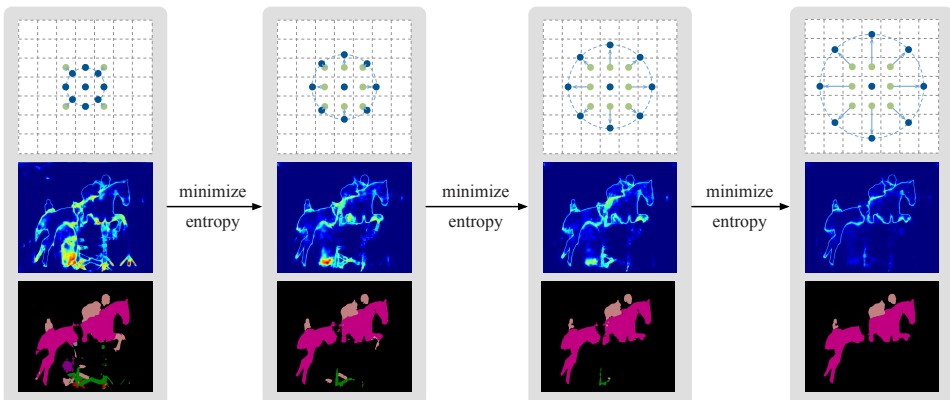

Figure 2: Overview. Dynamic receptive field scale (top) is optimized according to the output (bottom) at test time. We optimize receptive field scales and filter parameters to minimize the output entropy (middle). Optimizing during inference makes iterative updates shown from left to right: receptive field scale adapts, entropy is reduced, and accuracy is improved. This gives a modest refinement for training and testing at the same scale, and generalization improves for testing at different scales.

## 2.1 OBJECTIVE: ENTROPY MINIMIZATION

Unsupervised inference objectives can be bottom-up, based on the input, or top-down, based on the output. To augment already bottom-up prediction, we choose the top-down objective of entropy minimization. In essence, the objective is to reduce model uncertainty.

More precisely, for the pixel-wise output $\hat{Y} \in [0,1]^{C \times H \times W}$ for $C$ classes and an image of height $H$ and width $W$, we measure uncertainty by the Shannon entropy (Shannon, 1948):

$$\mathbf{H}_{i,j}(\hat{Y}) = -\sum_c \mathbf{P}(y_{i,j} = c) \log \mathbf{P}(y_{i,j} = c) \tag{1}$$

for each pixel at index $i, j$ to yield pixel-wise entropy of the same spatial dimensions as the output.

Entropy is theoretically motivated and empirically supported. By inspection, we see that networks tend to be confident on in-distribution data from the training regime. (Studying the probabilistic calibration of networks (Guo et al., 2017) confirms this.) In our case, this holds for testing scales similar to the training scales, with high entropy on segment contours. On out-of-distribution data, such as scale shifts, the output entropy is higher and less structured. For qualitative examples, see Figures 1 and 2.

This objective is severe, in that its optimum demands perfect certainty (that is, zero entropy). As a more stable alternative, we consider adaptively thresholding the objective by the average entropy across output pixels. We calculate the mean entropy at each iteration, and only take the gradient of pixels with above-average entropy. This mildly improves accuracy.

Our final objective is then:

$$L(\hat{Y}) = \sum_{i,j \in \mathbf{S}} \mathbf{H}_{i,j}(\hat{Y}) \text{ for } S = \{i,j : \mathbf{H}_{i,j} > \mathbf{H}_\mu\} \tag{2}$$

where $\mathbf{S}$ is the set of pixels with entropy above the average $\mathbf{H}_\mu$. At each step, we re-calculate the average entropy and re-select the set of violating pixels. In this way, optimization is focused on updating predictions where the model is the most uncertain.

## 2.2 VARIABLES: TASK AND STRUCTURE PARAMETERS

We need to pick the variables to optimize over so that there are enough degrees of freedom to adapt, but not so many that overfitting sets in. Furthermore, computation time and memory demand a minimal set of variables for efficiency. Choosing parameters in the deepest layers of the network satisfy these needs: capacity is constrained by keeping most of the model fixed, and computation is

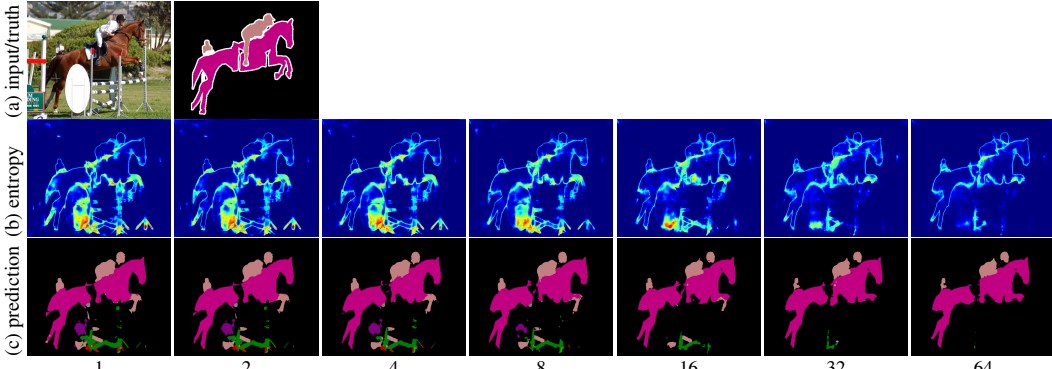

Figure 3: Iterative dynamic inference by our entropy minimization. We optimize output entropy with respect to task and scale parameters. (a) Input and ground truth. (b) Output entropy. (c) Output prediction. Our optimization reduces entropy and improves prediction accuracy.

reduced by only updating a few layers. The alternative of choosing all the parameters, and optimizing end-to-end during inference, is ineffective and inefficient: inference is slower and less accurate than feedforward prediction.

We select the task parameters $\theta_{\text{score}}$ of the output classification filter, for mapping from features to classes, and the structure parameters $\theta_{\text{scale}}$ of the scale regression filter, for mapping from features to receptive field scales. Optimizing over these parameters indirectly optimizes over the local predictions for classification scores $\hat{Y}$ and scales $\hat{\Sigma}$.

Why indirectly optimize the outputs and scales via these parameters, instead of direct optimization? First, dimensionality is reduced for regularization and efficiency: the parameters are shared across the local predictions for the input image and have fixed dimension. Additionally, this preserves dependence on the data: optimizing directly over the classification predictions admits degenerate solutions that are independent of the input.

### 2.3 Algorithm: Initialization, Iteration, and Termination

**Initialization** The unaltered forward pass of the base network gives scores $\hat{Y}^{(0)}$ and scales $\hat{\Sigma}^{(0)}$.

**Iteration** For each step $t$, the loss is the sum of thresholded entropies of the pixel-wise predictions $\hat{Y}^{(t)}$. The gradient of the loss is taken for the parameters $\theta_{\text{score}}^{(t)}$ and $\theta_{\text{scale}}^{(t)}$. The optimizer then updates both to yield $\theta_{\text{score}}^{(t+1)}$ and $\theta_{\text{scale}}^{(t+1)}$. Given the new parameters, a partial forward pass re-infers the local scales and predictions for $\hat{Y}^{(t+1)}$ and $\hat{\Sigma}^{(t+1)}$. This efficient computation is a small fraction of the initialization forward pass.

**Termination** The number of iterations is set and fixed to control the amount of inference computation. We do so for simplicity, but note that in principle convergence rules such as relative tolerance could be used with the loss, output, or parameter changes each iteration for further adaptivity.

Figure 3 shows the progress of our inference optimization across iterations.

## 3 Experiments

We experiment with extending from predictive to iterative dynamic inference for semantic segmentation, because this task has a high degree of appearance and scale variation. In particular, we show results for iterative optimization of classifier and scale parameters in a dynamic Gaussian receptive field model (Shelhamer et al., 2019) on the PASCAL VOC (Everingham et al., 2010) dataset. By adapting both task and structure parameters, our approach improves accuracy on in-distribution inputs and generalizes better on out-of-distribution scale shifts. We ablate which variables to optimize and for how many steps, and analyze our choices by oracle and adversary results. These experiments

establish the efficacy of entropy minimization during inference for scale adaptation, while oracle results show opportunity for further progress.

**Data and Metric** PASCAL VOC (Everingham et al., 2010) is a well-established semantic segmentation benchmark with 20 semantic classes and a background class. The original dataset only has 1,464, 1,449 and 1,456 images with segmentation annotations for training, validation, and testing, respectively. As is standard practice, we include the additional 9,118 images and annotations from Hariharan et al. (2011), giving 10,582 training samples in total. We measure accuracy by the usual metric of mean intersection-over-union (IoU). We report our results on the validation set.

**Architecture** We choose deep layer aggregation (DLA) (Yu et al., 2018) as a strong, representative fully convolutional network (Shelhamer et al., 2017) architecture. DLA exploits the feature pyramid inside the network via iterative and hierarchical aggregation across layers. We will release code and the reference models implemented in PyTorch (Paszke et al., 2017).

**Training** We train our model on the original scale of the dataset. We optimize via stochastic gradient descent (SGD) with batch size 64, initial learning rate 0.01, momentum 0.9, and weight decay 0.0001 for 500 epochs. We use the "poly" learning rate schedule (Chen et al., 2018) with power 0.9. For the model with no data augmentation ("w/o aug"), the input images are padded to $512 \times 512$ . As for the "w/ aug" model, data augmentation includes (1) cropping to $512 \times 512$, (2) scaling in $[0.5, 2]$, (3) rotation in $[-10°, 10°]$, (4) color distortion (Howard, 2013), and (5) horizontal flipping.

**Testing** We test our model on different scales of the dataset in the $[1.5, 4.0]$ range. We optimize the model parameters for adaptation via Adam (Kingma & Ba, 2015), batching all image pixels together, and setting the learning rate to 0.001. The model is optimized episodically to each input, and the parameters are reset between inputs. No data augmentation is used during inference to isolate the role of dynamic inference by the model.

## 3.1 RESULTS

We compare the semantic segmentation accuracy of our optimization with a prediction baseline and optimization by oracle and adversary. The baseline is a one-step dynamic model using feedforward scale regression to adapt receptive fields following (Shelhamer et al., 2019). We train on a narrow range of scales and test on a broader range of scales to measure refinement, the improvement for the training scales, and generalization, the improvement for the new scales. This baseline is the initialization for our iterative optimization approach: the output and scale predictions for the initial iteration are inferred by the one-step model. For analysis results, the oracle and adversary optimize during inference to respectively minimize/maximize the cross-entropy loss of the output and the truth.

As reported in Table 1, our method consistently improves on the baseline by $\sim$2 points for all scales, which indicates that our unsupervised optimization for iterative inference helps the model generalize better across scales. When the scale shift is larger, there is likewise a larger gap.

To evaluate the effect of data augmentation, we experiment with ("w/ aug") and without ("w/o aug"). Data augmentation significantly improves generalization across scales. Note that our optimization during inference still improves the model with data augmentation by the same amount.

|  |  | 1.5× | 2.0× | 2.5× | 3.0× | 3.5× | 4× |
|---|---|---|---|---|---|---|---|
| | scale regression | 68.2 | 59.3 | 50.2 | 41.8 | 34.0 | 27.5 |
| w/o aug | entropy optimization (ours) | **69.0** | **60.1** | **51.9** | **43.5** | **35.8** | **29.2** |
| | oracle | 72.0 | 64.4 | 55.8 | 47.5 | 39.2 | 32.1 |
| | scale regression | 74.2 | 70.8 | 65.8 | 59.8 | 53.5 | 46.8 |
| w/ aug | entropy optimization (ours) | **74.6** | **71.7** | **67.7** | **61.8** | **56.0** | **49.0** |
| | oracle | 78.0 | 75.7 | 72.3 | 67.8 | 62.4 | 55.6 |

Table 1: Comparison of our method with the feedforward scale regression baseline and the oracle. Results are scored by intersection-over-union (higher is better). "w/o aug" excludes data augmentation, where "w/ aug" includes scaling, rotation, and other augmentation. Even though data augmentation reduces the effect of scale variation, our method further improves accuracy for all scales.

| | | 1.5× | 2.0× | 2.5× | 3.0× | 3.5× | 4× |
|---|---|---|---|---|---|---|---|
| step 0 | scale regression | 68.2 | 59.3 | 50.2 | 41.8 | 34.0 | 27.5 |
| step 32 | entropy optimization (ours) | **69.0** | 60.1 | 51.9 | **43.5** | **35.8** | **29.2** |
| | oracle | 72.0 | 64.4 | 55.8 | 47.5 | 39.2 | 32.1 |
| step 128 | entropy optimization (ours) | 69.0 | **60.3** | **52.1** | **43.5** | 35.2 | 28.5 |
| | oracle | 73.3 | 68.6 | 61.8 | 54.0 | 45.7 | 38.5 |

Table 2: Ablation of the number of iterations: entropy minimization saturates after 32 steps.

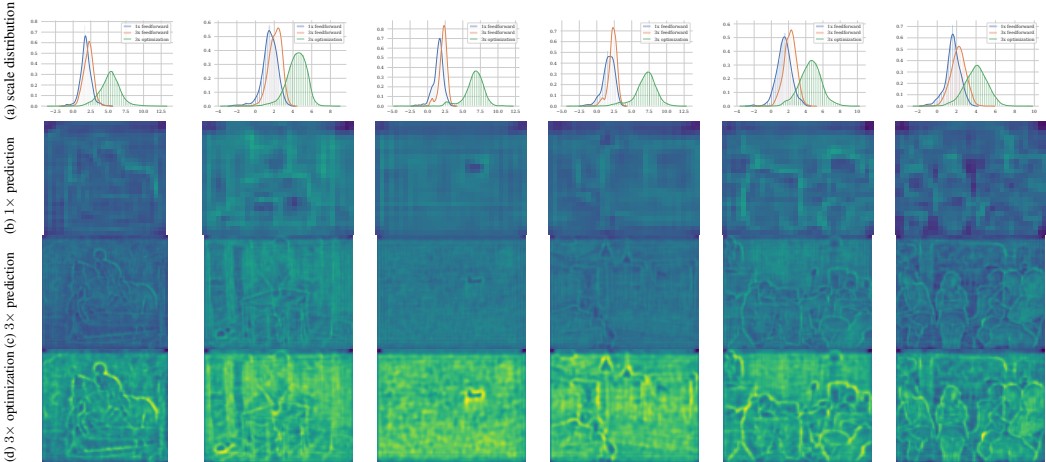

Figure 4: Analysis of dynamic receptive field sizes across scale shift. (a) plots the distribution of dynamic receptive fields, confirming that optimization shifts the distribution further. (b) is the prediction at 1× scale while (c) and (d) are the prediction baseline and our iterative optimization at 3× scale. (c) and (d) are visually similar, in spite of the 3× shift, showing that the predictor has failed to adapt. Optimization adapts further by updating the output and scale parameters, and the receptive fields are accordingly larger. For visualization darker indicates smaller, and brighter indicates larger.

## 3.2 ABLATIONS

We ablate the choice of parameters to optimize and the number of updates to make.

We optimize during inference to adapt the task parameters (score) of the classifier and structure parameters (scale) of the scale regressor. The task parameters map between the visual features and the classification outputs. Updates to the task parameters are the most direct way to alter the pixelwise output distributions. Updates to the structure parameters address scale differences by adjusting receptive fields past the limits of the feedforward scale regressor. From the experiments in Table 3, both are helpful for refining accuracy and reducing the generalization gap between different scales. Optimizing end-to-end, over all parameters, fails to achieve better than baseline results.

Iterative optimization gives a simple control over the amount of computation: the number of updates. This is a trade-off, because enough updates are needed for adaptation, but too many requires excessive computation. Table 2 shows that 32 steps are enough for improvement without too much computation. Therefore, we set the number of steps as 32 for all experiments in this paper. For our network, one step of inference optimization takes $\sim \frac{1}{10}$ the time of a full forward pass.

## 3.3 ANALYSIS

We analyze the distribution of scales in Figure 4 and show qualitative segmentation results in Figure 5.

While better compensating for scale shift is our main goal, our method also refines inference on in-distribution data. The results in Table 3 for 1× training and testing show improvement of ∼1 point.

We analyze our approach from an adversarial perspective by maximizing the entropy instead of minimizing. To measure the importance of a parameter, we consider how much accuracy degrades

when adversarially optimizing it. The more performance degrades, the more it matters. Table 3 shows that adversarial optimization of the structure parameters for scale degrades accuracy significantly, indicating the importance of dynamic scale inference. Jointly optimizing over the task parameters for classification further degrades accuracy.

| | test on $1\times$ | | | test on $3\times$ | | |
|---|---|---|---|---|---|---|
| | score | scale | both | score | scale | both |
| scale regression | 69.8 | 69.8 | 69.8 | 59.8 | 59.8 | 59.8 |
| entropy optimization (ours) | 70.2 | **70.7** | **70.6** | 61.1 | 61.8 | **62.3** |
| oracle | 73.7 | 75.6 | 77.7 | 63.9 | 67.8 | 71.3 |
| adversary | 67.4 | 55.9 | 52.4 | 57.4 | 47.4 | 44.4 |

Table 3: Analysis of entropy minimization (compared to oracle and adversary optimization) and ablation of the choice of parameters for optimization (score, scale, or both). The oracle/adversary optimizations minimize/maximize the cross-entropy of the output and truth to establish accuracy bounds. The adversary results show that our method helps in spite of the risk of harm. The oracle results show there are still better scales to be reached by further progress on dynamic inference.

## 4 RELATED WORK

**Dynamic Inference** Dynamic inference adapts the model to each input (Olshausen et al., 1993). Many approaches, designed (Lindeberg, 1994; Lowe, 2004) and learned (Jaderberg et al., 2015; De Brabandere et al., 2016; Dai et al., 2017; Perez et al., 2017; Shelhamer et al., 2019), rely on bottom-up prediction from the input. Our method extends bottom-up prediction with top-down optimization to iteratively update the model from the output. Recurrent approaches to iterative inference (Pinheiro & Collobert, 2014; Carreira et al., 2016) require changing the architecture and training more parameters. Our optimization updates parameters without architectural alteration.

**Entropy Objective** We minimize entropy during testing, not training, in effect tuning a different model to each input. The entropy objectives in existing work are optimized during training, especially for regularization. Entropy is maximized/minimized for domain adaptation (Tzeng et al., 2015; Long et al., 2016; Vu et al., 2018; Saito et al., 2019) and semi-supervised learning (Grandvalet & Bengio, 2005; Springenberg, 2016). In reinforcement learning, maximum entropy regularization improves policy optimization (Williams & Peng, 1991; Ahmed et al., 2019). We optimize entropy locally for each input during testing, while existing use cases optimize globally for a dataset during training.

**Optimization for Inference** We optimize an unsupervised objective on output statistics to update model parameters for each test input. Energy minimization models (LeCun et al., 2006) and probabilistic graphical models (Koller & Friedman, 2009; Wainwright & Jordan, 2008) learn model parameters during training then optimize over outputs during inference. The parameters of deep energy models (Belanger et al., 2017; Gygli et al., 2017) and graphical models are fixed during testing, while our model is further optimized on the test distribution. Alternative schemes for learning during testing, like transduction and meta-learning, differ in their requirements. Transductive learning (Vapnik, 1998; Joachims, 1999) optimizes jointly over the training and testing sets, which can be impractical at deep learning scale. We optimize over each test input independently, hence scalably, without sustained need for the (potentially massive) training set. Meta-learning by gradients (Finn et al., 2017) updates model parameters during inference, but requires supervision during testing and more costly optimization during meta-training.

## 5 CONCLUSION

Dynamic inference by optimization iteratively adapts the model to each input. Our results show that optimization to minimize entropy with respect to score and scale parameters extends adaptivity for semantic segmentation beyond feedforward dynamic inference. Generalization improves when the training and testing scales differ substantially, and modest refinement is achieved even when the training and testing scales are the same. While we focus on entropy minimization and scale inference, more optimization for dynamic inference schemes are potentially possible through the choice of objective and variables.

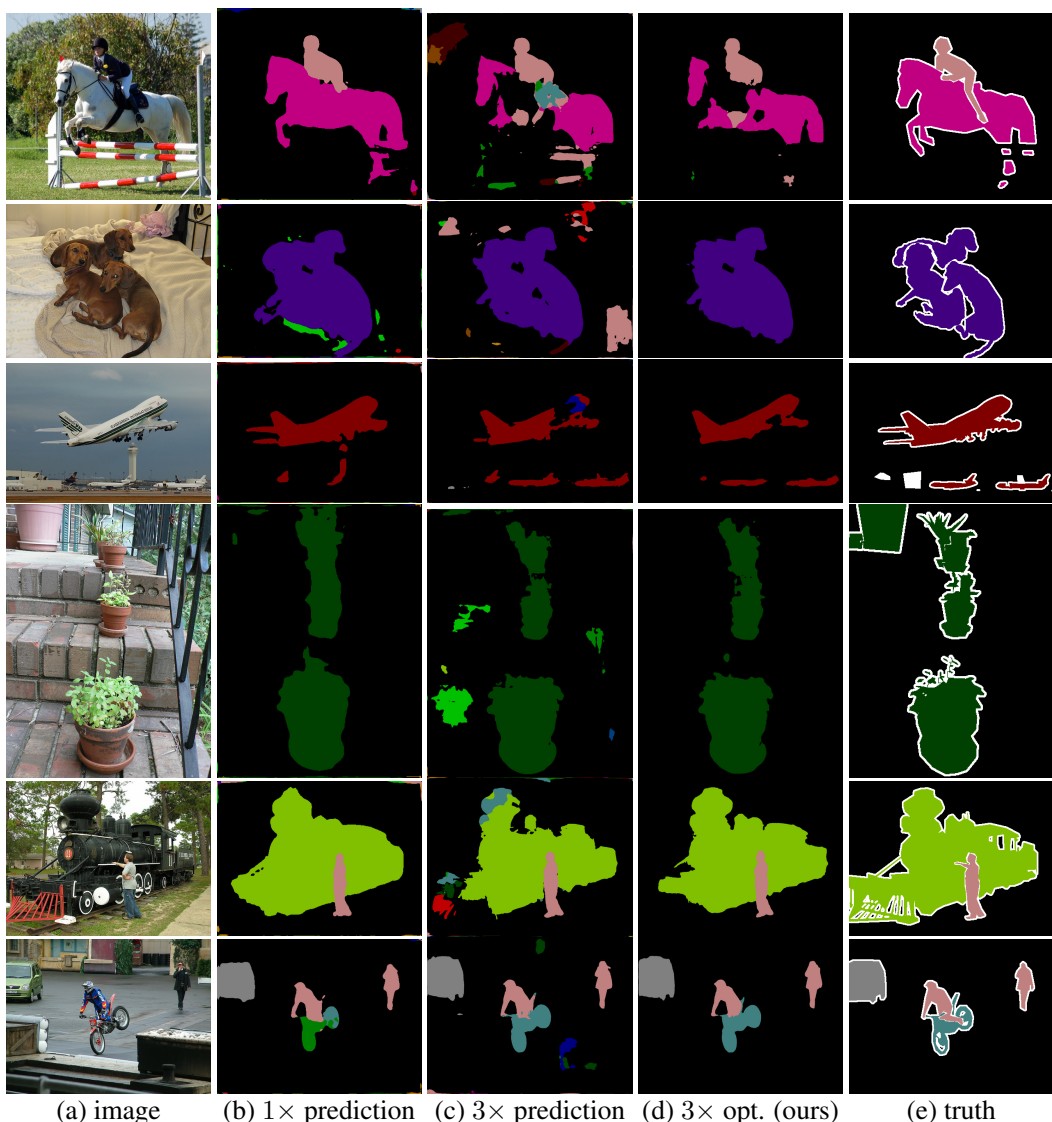

(a) image    (b) 1× prediction    (c) 3× prediction    (d) 3× opt. (ours)    (e) truth

Figure 5: Qualitative results from the PASCAL VOC validation set. Our model is trained on 1× scale and tested on 3× scale. (a) and (e) are the input image and ground truth. (b) indicates the reference in-distribution prediction on 1× scale. (c) is the out-of-distribution prediction for the dynamic prediction baseline. (d) is the out-of-distribution prediction for our iterative optimization method. Our method corrects noisy, over-segmented fragments and false negatives in true segments.

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
