# OpenReview forum: "Dynamic Scale Inference by Entropy Minimization"
_ICLR.cc/2020/Conference — Reject_

### Official Review · AnonReviewer3 · 2019-10-23
**Official Blind Review #3**

**Rating:** 3

**Review:**

This paper focused on the problem of semantic segmentation. The author proposed to minimize the output entropy to dynamically predict the scales when doing inference. The entropy minimization strategy is achieved by iterative optimization. Experimental results are reported on the PASCAL VOC dataset.

Clarity:
I think this paper is moderate. The idea of dynamically predicting the scale or receptive field is interesting. However, this issue can be addressed through multi-scale training/testing or the deformable kernels. The experimental results are not that convincing. The method is only evaluated on one dataset and one backbone. The paper could be improved with more convincing experiments.

Limitations:
The optimization process may take a certain number of forward and backward steps. In Sec 3.2 the author shows this will introduce 3x more time, this will much decrease its popularity when compared to other scale processing methods like deformable kernels.

Experiments:
1. The proposed method is evaluated on the PASCAL VOC dataset with the DLA segmentation backbone. The chosen backbone is not as strong as the most popular frameworks like DeepLab and PSPNet. Thus the baseline results as shown in Table 1 are not that high. I would like to see the relative improvements introduced by the proposed method over a stronger baseline.

2. The experimental dataset is PASCAL VOC only. I would be more convinced with more datasets like Cityscapes or ADE20K.

3. The reported experimental results are with models trained on a narrow range of scales. What the results and relative improvements would be if trained with regular multi-scales like [0.5, 2.0]? Will the scale issue be easily addressed by a multi-scales training strategy?

4. The number of optimization steps may be hard to control, 32 is used for DLA on PASCAL VOC. Will this number be changed for different models on different datasets? If yes, can the author find a more elegant way to decide when to end the optimization process?

Misc:
It is better to give a brief introduction of structure parameters scale and dynamic Gaussian receptive fields as in Sec 2.3.

**Experience Assessment:**

I have published in this field for several years.

**Review Assessment: Checking Correctness Of Derivations And Theory:**

I carefully checked the derivations and theory.

**Review Assessment: Checking Correctness Of Experiments:**

I carefully checked the experiments.

**Review Assessment: Thoroughness In Paper Reading:**

I read the paper thoroughly.

---

> ### Author Response · Authors · 2019-11-15
> **Multi-scale Training and Deformation are Included and Improved On (Plus: How to End Optimization)**
>
> Thank you for your feedback and consideration of alternatives for addressing scale.
>
> > this issue can be addressed through multi-scale training/testing or the deformable kernels
>
> We do multi-scale training with data augmentation (see "w/ aug" results). The improvement by optimization during inference is on top of a base model trained this way. Multi-scale testing requires knowledge of the testing scale, or enumeration of several scales, while the purpose of our method is to select the unknown scale.
>
> The base method "scale regression" does include deformable kernels, in particular scale deformations with Gaussian structure (Shelhamer et al. 2019). The improvement due to optimization is on top of this adaptation by prediction, showing the need for further adaptation by optimization. Please see Figure 4 for the comparison of scale deformation by prediction and optimization.
>
> > optimization process may take [more time] when compared to other scale processing methods like deformable kernels
>
> We appreciate the accuracy and efficiency of deformable kernels. Our experiments show that they are however limited, in that the predictor for the deformation can only generalize so far. Optimization takes more time, but it does so to improve the results.
>
> Note that multi-scale testing also takes more time. Since computation scales as the product of input dimensions,, a 2x larger input would cost ~4x the time, and be comparable or more expensive than our optimization.
>
> > The number of optimization steps may be hard to control
> > [is there] a more elegant way to decide when to end the optimization process?
>
> We agree that a more adaptive rule for ending optimization would be preferable. In a rebuttal experiment on a simpler ResNet-50 FCN, we achieved similar results by choosing the number of iterations or choosing a relative tolerance on the change in the entropy across iterations. This tolerance could be more transferrable than a fixed number of steps.

---

### Official Review · AnonReviewer2 · 2019-10-23
**Official Blind Review #2**

**Rating:** 6

**Review:**

The authors propose a method to dynamically adapt some structural features of a semantic image segmentation model at inference time based on the entropy of the predictions.

Using a model that explicitly controls the size of the filters at each layer, they show that running a small number of SGD steps on the scale and final prediction parameters in the last layer to minimize the entropy of least confident predictions for a specific example leads to better performance overall, and especially better generalization when there is a size discrepancy between training and test set.

Strengths: The method is inspired, and leads to significant improvements. The dynamic inference setup is clearly explained, and well motivated for the case of the scale parameters. Extensive ablation experiments and the inclusion of an oracle system help understand the contributions of each component of the setup, and the potential of inference-time optimization of the considered parameters.

Weaknesses: Some information is missing from the description of the experimental setting. A quick review of the DLA model would be welcome, to get a better sense of the roles of \theta_{scale} and \theta_{score}. The authors should include published numbers for a relevant baseline and the current or recent state-of-the-art for the considered dataset (Table 1should also report 1x numbers in both settings). Finally, while the authors make a strong case for dynamic adaptation of the scale parameters, the prior reason for adapting \theta_{score} is less obvious and would require further explanation.

Questions and miscellaneous remarks:
“As reported in Table 1, our method consistently improves on the baseline by ∼2 points for all scales” > this statement is a little misleading, since the improvement is ~2 points on average, not ~2 points for each scale.

Wouldn’t simply multiplying \theta_{score} by a large number decrease the entropy of the predictions? Do you do anything to prevent that from happening?

Similarly, couldn’t the adversary simply rotate \theta_{score} to reduce IoU? Is the adversary optimized for long enough?


**Experience Assessment:**

I have read many papers in this area.

**Review Assessment: Checking Correctness Of Derivations And Theory:**

I assessed the sensibility of the derivations and theory.

**Review Assessment: Checking Correctness Of Experiments:**

I assessed the sensibility of the experiments.

**Review Assessment: Thoroughness In Paper Reading:**

I read the paper thoroughly.

---

> ### Author Response · Authors · 2019-11-15
> **Optimizing Scale and Score, State-of-the-Art, and Simple Parameter Transformations**
>
> Thank you for the feedback and careful consideration of the optimization problem.
>
> > the prior reason for adapting \theta_{score} is less obvious and would require further explanation
>
> For completeness, we try adapting both scale and score. The purpose of adapting score is to further fit the model to the appearance of each test input. This can be seen as a way to add modeling capacity, since every test input has its own classifier parameters, instead of sharing the same parameters across all inputs.
>
> > current or recent state-of-the-art for the considered dataset
>
> The current state-of-the-art for PASCAL VOC is DeepLabv3+ [1] at a commanding 89% IoU. Many factors contribute to its high accuracy, including supervision, optimization, and inference-time augmentation that are independent of scale.
>
> We choose DLA for the relevance of its skip connections and multi-scale feature pyramid, to show that further scale adaptation still helps.
>
> > Wouldn’t simply multiplying \theta_{score} by a large number decrease the entropy
> > Similarly, couldn’t the adversary simply rotate \theta_{score} to reduce IoU?
>
> While there are simple parameter transformations to decrease entropy or reduce accuracy, that does not mean they are simple to optimize, and empirically these cases do not happen. We do not regularize by weight decay during inference to keep from multiplying the score parameters. We do optimize the adversary until the loss converges (but note that IoU might not drop to zero becauses ties are broken by class order, with background first, so if the parameters are driven to zero the output will be fully background and be partially correct).
>
> > improvement is ~2 points on average, not ~2 points for each scale
>
> Thank you for your precision. We will rephrase this accordingly.
>
> [1] https://arxiv.org/abs/1802.02611

---

### Official Review · AnonReviewer4 · 2019-10-30
**Official Blind Review #4**

**Rating:** 3

**Review:**

Summary:
The following work proposes a test-time optimization over scales to improve semantic segmentation. Specifically, at test time, they iteratively optimize over the score and scale parameters of Shellhamer et al 2019, where a Gaussian receptive field is used to allow for dynamic scale adaptation of each convolutional layer. They optimize the parameters with respect to an entropy minimization objective. Experiments on PASCAL VOC, reported at multiple scales, demonstrate improvements in IOU over the vanilla architecture.

Strengths:
-The work was well-motivated
-Formulation is pretty elegant and outperforms the baseline

Weaknesses:
While I liked the somewhat elegant formulation of dynamic test-time scaling proposed by the following work, I don't think this work introduced many novel results nor insights
-Multiscale test-time inference is already standard in state-of-the-art architectures such as DeepLab. Specifically, DeepLab runs inference at multiple scales, then max pools the logit responses across all scales. (see Table 3 of https://arxiv.org/pdf/1511.03339.pdf)
-Using entropy as a measure of network uncertainty is a good idea, but also not a novel finding.
-COCO and Cityscapes would probably have been better choices for datasets with larger variations in scale

Improvements:
-Try comparing against the multiscale logit-max-pooling inference procedure as a baseline -- or demonstrate that the proposed technique can further improve upon the results of the logit-pooling technique.
-Some details weren't very well explained. Specifically, what is the $\theta_{score}$ task parameter for?

** Post Rebuttal Response
I'd like to thank the authors for clarifying some points in their response. Overall, I maintain that I think the optimization-based scale inference solution they present is interesting from an implementation standpoint, but the findings in this work did not yield sufficient new insight for the task. While I agree that their approach is fairly different from common approaches such as discrete image pyramids, a thorough quantitative comparison of these differences would make this work a lot stronger. As such, I maintain my original rating.

**Experience Assessment:**

I have published one or two papers in this area.

**Review Assessment: Checking Correctness Of Derivations And Theory:**

N/A

**Review Assessment: Checking Correctness Of Experiments:**

I assessed the sensibility of the experiments.

**Review Assessment: Thoroughness In Paper Reading:**

I read the paper thoroughly.

---

> ### Author Response · Authors · 2019-11-15
> **Multi-scale Testing, Novelty of Entropy for Inference-Time Optimization, and Task Parameters**
>
> Thank you for your feedback and careful consideration of inference across scales.
>
> > Multiscale test-time inference is already standard in state-of-the-art architectures
>
> The choice of enumeration by pyramid or selection by adaption is both practical and philosophical. Practically speaking, pyramids are common for deep learning in vision, but selection is not, so a point of this work is to show that selection (especially by optimization) is worthy of more attention. In the paper referenced by the review, Table 3 shows a 1-2 point gain by pyramid, which is comparable to our 2 point gain by a substantially different route. More philosophically, a pyramid is discrete and fixed while our optimization is continuous and adaptive.
>
> > Using entropy as a measure of network uncertainty is a good idea, but also not a novel finding
>
> Entropy is certainly a well-established measure of uncertainty. The novelty of our method is its use for optimization during inference: to the best of our knowledge, ours is the first method to adapt to each testing input by entropy minimization.
>
> > what is the task parameter for?
>
> The task parameters are the parameters of the output layer (also known as the "score" layer). By optimizing over scale and score parameters, the method can adjust geometry and appearance to minimize entropy. In Table 3 we evaluate optimizing only scale, only score, or both and find that both can help.

---

### Decision · Program_Chairs · 2019-12-19

**Decision:**

Reject

**Comment:**

This paper constitutes interesting progress on an important problem.  I urge the authors to continue to refine their investigations, with the help of the reviewer comments; e.g., the quantitative analysis recommended by AnonReviewer4.